

# Performance and sensitivity of column-wise and pixel-wise methane retrievals for imaging spectrometers

Alana K Ayasse[1], Daniel Cusworth[1], Kelly O'Neill[1], Justin Fisk[1], Andrew K Thorpe[2], Riley Duren[1,2,3]

1Carbon Mapper Inc., Pasadena, CA 91101, USA
2Jet Propulsion Laboratory, California Institute of Technology, Pasadena, CA 91109, USA
3University of Arizona, Tucson, AZ 85721, USA

*Correspondence to*: Alana K Ayasse (alana@carbonmapper.org)

**Abstract.** Strong methane point source emissions generate large atmospheric concentrations that can be detected and quantified with infrared remote sensing and retrieval algorithms. Two standard and widely used retrieval algorithms for one class of observing platform, imaging spectrometers, include pixel-wise and column-wise approaches. In this study, we assess the performance of both approaches using the airborne imaging spectrometer (Global Airborne Observatory) observations of two extensive controlled release experiments. We find that the column-wise retrieval algorithm is sensitive to the flight line length and can have a systematic low bias with short flight lines, which is not present in the pixel-wise retrieval algorithm. However, when the flight line length is sufficiently long, the column-wise retrieval algorithms produce results that very closely match metered emission rates. Lastly, this study examines the methane plume detection performance of GAO with a column-wise retrieval algorithm, and finds minimum detection limits between 9-10 kg/hr and 90% probability of detection between 10-45 kg/h. These results present a framework of rules for guiding proper concentration retrieval selection given conditions at the time of observation to ensure robust detection and quantification.

## 1 Introduction

Multiple studies have shown that in many regions across emission sectors, a significant component of the anthropogenic methane (CH4) budget come from a relatively small population of high emission discrete point sources (Lyon et al., 2015; Lauvaux et al., 2022; Sherwin et al., 2023, Duren et al 2019, Cusworth et al., 2022). This result has significant policy implications because identifying and mitigating a large proportion of CH4 emissions quickly is required to limit adverse climate warming effects in the next few decades. Several atmospheric remote sensing platforms are particularly sensitive to these emission types. In particular, airborne imaging spectrometers with shortwave infrared (SWIR) sensitivity have emerged as useful tools for point source quantification due to their high spatial resolution and ability to repeatedly map large areas for intermittent and/or stochastic point sources. The accuracy of emissions quantification of point sources depends on a combination of instrument performance, the methane concentration retrieval algorithm, plume identification and delineation, and environmental variables including surface illumination and atmospheric transport. Methane retrieval algorithms for remote sensing platforms that rely on solar backscattered radiance, like imaging spectrometers, vary in implementation and



complexity. As the ecosystem of airborne and satellite imaging spectrometers grows, understanding how retrieval assumptions propagate to emission estimates is required to ensure remote sensing derived emission estimates are robust and accurate.

Controlled release experiments provide a means of independently evaluating detection limits and uncertainty in emission estimates. Initial unblinded controlled release experiments with the next generation Airborne Visible/Infrared Imaging Spectrometer (AVIRIS-NG) were performed to assess detection limits and quantification accuracy at relatively low emission rates (max release was 141 kg/hr) (Thorpe et al., 2016). In 2021 and 2022 Stanford University conducted more comprehensive, blinded controlled release experiments to validate multiple ground-based, airborne, and satellite CH4 sensing technologies.
Carbon Mapper participated in both experiments, using the Global Airborne Observatory (GAO) imaging spectrometer, which has the same design as AVIRIS-NG. Carbon Mapper flights resulted in over 250 observations of metered emission rates between 5 and 1500 kg/hr. These data provide an excellent opportunity to test, compare, and validate methane emissions retrieval algorithms.

In this paper we use now unblinded controlled release data to provide a quantitative sensitivity assessment of two methane retrieval algorithms: column-wise matched filter and pixel-wise Iterative Maximum A Posteriori – Differential Optical Absorption Spectroscopy (IMAP-DOAS) algorithm. We also use these data to assess a minimum detection limit for the Carbon Mapper airborne platform. During the 2021 and 2022 controlled release experiments, we employed two observing strategies with GAO: (1) rapid repeat surveys focused on the controlled release site, which resulted in many observations but smaller
image sizes (2021 experiment), and (2) broad imaging of the controlled release site and surrounding areas, which resulted in greater characterization of the background but fewer observations (2022 experiment). The latter strategy was more representative of our standard operations for mapping large regions. We show that performance of retrieval algorithm is strongly sensitive to observing strategy, specifically for column-wise retrievals that depend on background characterization using many pixels across the scene. These results establish a general rule framework for selecting retrievals and quantifying
systematic biases based on observing conditions at time of acquisition. These results corroborate a 10 kg/hr detection limit for this class of airborne imaging spectrometer, but also highlight the complexities and contributing factors that alter detection limits on a scene-by-scene bases. The rules and frameworks established here can be applied and adapted to other observing platforms that may apply similar CH4 retrieval approaches.

## 2 Methods

### 2.1 Controlled Release Experiments

A series of controlled release experiments were performed by a Stanford University team in summer 2021 and fall 2022 (Rutherford et al., 2023; Sherwin et al., 2023; El Abbadi et al., 2023). These experiments evaluated the detection limits and



quantification accuracy of various CH4 measurement technologies, including ground-based, airborne, and satellite platforms. In 2021, Carbon Mapper participated in controlled release tests conducted on July 30, 31, and Aug 3, 2021 near Midland,

Texas. The metered release rates were between 10 and 1500 kg/hr. In addition, a sonic anemometer was located at the site to provide wind speed and direction data at 10 meters above ground level. In 2022, Carbon Mapper participated in controlled release tests on Oct 10-12, 28, 29, and 31, 2022 near Casa Grande, Arizona. The metered release rates were between 5 and 1450 kg/hr and 10 meter sonic anemometer data was also provided. For more details on the controlled release experiments see Rutherford et al., 2023 (2021 experiments) and El Abbadi et al., in prep (2022 experiments).


For both 2021 and 2022 controlled releases, we used the GAO platform. The Visible/Short-Wave Infrared (VSWIR) imaging spectrometer onboard GAO measures ground-reflected solar radiation at wavelengths from 380 nm to 2510 nm with 5-nm spectral sampling. The instrument spatial resolution is correlated to the flight altitude. For these studies the instrument was flown at 10,000 ft (~3km) resulting in a roughly 3 m pixel size. GAO has a 34 degree field of view, which results in an

approximate swath width (or cross-track extent) of approximately 2 km when flown at 3 km altitude. Flight line length refers to the along-track direction of data acquisition and can vary in length depending on observing preferences.

In the first controlled release experiment, we maximized the number of observations during the short campaign window by flying short (~3km) flight lines around the controlled release site. This resulted in 229 observations of the controlled release

site (including null releases). However, this flightline length was well below our standard observing practice and had consequences on the column-wise retrieval, so for the second controlled release experiment the flightline length was extended to 20 km, more representative of normal survey operations, which resulted in 121 observations of the controlled release site. In both experiments we applied our standard quality control protocols to eliminate scenes with clouds or cloud shadows, unstable plume morphology, or unstable wind conditions. After quality checks, there were 163 plumes for the 2021 controlled

release experiment and 87 plumes for the 2022 experiment.

**2.2 Column-Wise Retrieval**

The column matched filter (CMF) is a column wise statistical algorithm to estimate CH4 concentration enhancements. CMF algorithms are used operationally for Carbon Mapper airborne campaigns given their computational efficiency and ability to reduce systematic instrument error that may occur across nonuniformly across-track elements in an imaging spectrometer. The

algorithm takes the following form (Thompson et al. 2015):

$$\hat{\alpha}(x) \ = \ (x - \mu)^T \Sigma^{-1} t \ /(t^T \Sigma^{-1} t ) \tag{1}$$

Where $\hat{\alpha}$ is the path-length concentration CH4 enhancement (units ppm-m), $x$ is a radiance spectrum, $\mu$ is the mean radiance

spectrum in an along-track column, $\Sigma$ is a covariance matrix, and $t$ is a unit absorption spectrum. Vectors **x, $\mu$,** and **$t$** include



71 elements, which represents all bands between [2104, 2459] where CH4 has known absorption properties. In essence, the matched filter uses statistics from all pixels in a flight column to assess whether a

a single pixel's spectrum is enhanced by methane (i.e., has deeper absorption features). Sufficient column-wise statistics (i.e., pixels) are needed to define a robust covariance matrix.

Initially, we supposed that the number of pixels in a column (i.e., flight line length) should be at least seven times larger the number of active bands used in the retrieval, and that no columns should have more than 5% of its pixels enhanced by methane. Since we use 71 active bands in the retrieval, this suggests a ~497 pixel flight line length minimum (1.5 km for a flight altitude of 3 km). Though this reasoning was used in general to define minimum flight line lengths, in practice, Carbon Mapper performs wide area surveys of many facilities on regional and basin scales, which result in much longer flight line lengths.

**2.3 Pixel-Wise Retrieval**

The IMAP-DOAS algorithm is a pixel-wise methane retrieval. IMAP-DOAS estimates dry air column-average methane concentrations (XCH4; units ppb) on a per-pixel basis by simulation of top of the atmosphere radiance and inversion (or retrieval) for the best atmospheric parameters that reduce mismatch between an observed spectrum and a simulated spectrum, assuming some prior constraints. For the simulated spectrum, IMAP-DOAS uses a radiative transfer model that relies on a

multi-layered Beer-Lambert Law equation to simulate high frequency atmospheric features and a multi-dimensional polynomial to represent low-frequency reflectance and scattering features (Cusworth et al. 2019; C. Frankenberg, Platt, and Wagner 2005; Andrew K. Thorpe et al. 2017). The simulated spectrum must account for H2O and N2O, the other absorbing gases in the 2210-2400 nm spectra window, and low frequency surface features modelled as polynomial of order k = [0,k]. Therefore the state vector(x) is composed of the following :


$$x = (s_{CH4}, s_{H20}, s_{N2O}, a_0, ..., a_k) \tag{2}$$

Where s is a scaling factor applied to the column mixing ratio for each gas from the US Standard Atmosphere. To retrieve the state vector from the radiance we apply a forward model:


$$F^h(x) = I_0(\lambda)exp(-A \sum_{n=1}^{3} s_n \sum_{l=1}^{72} \tau_{n,l}) \sum_{k=0}^{K} \alpha_k P_k(\lambda) \tag{3}$$

Where $F^h$ is the high resolution TOA radiance at Wavelength $\lambda$, $I_0$ is the solar spectrum, $A$ is the geometric air mass factor, $\tau_{n,l}$ is the optical depth for optical depths for either CH4, H2O, or N2O (n) and the vertical level (l), $s_n$ is the scaling factor for

that optical depth, $\alpha$ is a polynomial coefficient, and P is the kth polynomial. The optical depth $\tau_{n,l}$ is calculated at each wavelength by multiplying the HITRAN absorption cross section by the volume mixing ratio (VMR) and the vertical column density of dry air (VCD) in a 72-layered atmosphere from the MERRA-2 meteorological reanalysis.



In order to model the TOA radiance we take $F^h(X)$ over the 2210-2410 nm spectral range at 0.02 nm resolution and convolve the spectrum using the band centers and FWHM from the instrument (for AVIRIS-NG this is a 5 nm spacing and a 6 nm FWHM). The observed TOA radiance (y) is represented as:

$$y = F(x) + \varepsilon \tag{4}$$

Where $\varepsilon$ is the observational error. The forward model is non-linear so the solution must be obtained iteratively. At each iteration ($i$) a Jacobian matrix is calculated of the state vector.

$$K_i = \frac{\partial F}{\partial x}\big|_{x = x_i} \tag{5}$$

We use Gauss-Newton iteration to solve for the optimal state vector:

$$
\begin{aligned}
x_{i+1} = \ & x_A + (K_i^T S_O^{-1} K_i + S_A^{-1})^{-1} K_i^T S_A^{-1} \\
& [y - F(x_i) + K_i(x_i - x_A)]
\end{aligned}
\tag{6}
$$

$S_O$ is the error covariance matrix defined by the SNR, $x_A$ is the prior estimate of the state vector, and $S_A$ is the prior error covariance matrix.

IMAP-DOAS has been used in multiple previous studies for a smaller population of emission sources (Cusworth et al. 2020; Andrew K. Thorpe et al. 2017; Cusworth et al. 2021) but is currently not run operationally for larger area surveys due to computational constraints. The benefit of IMAP-DOAS is that (1) each retrieved pixel is independent and (2) that retrieval uncertainties can be explicitly characterized by the Bayesian formulation of the retrieval. This contrasts with the CMF approach, where the retrieved concentration for a single pixel depends on the quality and density of pixels in an acquisition (e.g., a single along-track column).

## 2.4 Emission Rates

For both CMF and IMAP-DOAS, we estimate emission rates via the Integrated Methane Enhancement (IME) method (Christian Frankenberg et al. 2016; Varon et al. 2018):

$$Q = \frac{IME}{L} * U_{eff} \tag{7}$$



Where $Q$ is the emissions rate, *IME* is the integrated mass enhancement in kilograms, $L$ is the length in meters, in this case the length is calculated using the square root of the area of the plume. $U_{eff}$ is the effective wind speed and is calculated from the 10 meter anemometer winds using the following equation:

$$U_{eff} = 1.1 * log(U) + 0.6 \qquad (8)$$

Where $U$ is the 10 meter anemometer wind speed. In the absence of a 10 meter anemometer wind observation at the site of the plume, HRRR reanalysis products are used to estimate the 10 meter wind speed at the time and location of the observed methane plume.

The IME was calculated slightly differently for IMAP-DOAS and the CMF. For the CMF (which retrieves an enhancement above background) we calculated thresholds using the 80-98[th] percentile of 1 km area around the plume origin. The thresholds were then used to filter out low values, we then calculated connected components starting at the plume origin, then used a dilation to fill in gaps. The resulting plume mask was used to calculate the IME and L, the IME and L values calculated at each threshold were averaged together to get the final IME and L. For IMAP-DOAS (which retrieves a total column concentration) we used similar methods but first subtracted off the background concentration before calculating the IME. The background concentration was determined by taking the pixels not included in the plume mask and taking a percentile ( 95[th] percentile for the 2021 plume and 99[th] percentile for the 2022 plumes). For the matched filter emission results the uncertainties were derived from the standard deviation of the wind speeds 90 seconds prior to the observations and from the standard deviation of the IME and L due to different thresholds. For the IMAP-DOAS results the uncertainties are derived directly from the retrieval and from the standard deviation of the winds as stated above.

## 3 Results and Discussion

### 3.1 2021 Controlled Release

Estimated airborne emission rates compared to metered emissions are shown in Figure 1 for both CMF and IMAP-DOAS. An ordinary least squares (OLS) fit between the CMF results and the metered emissions results in y = 0.26x + 146 and R² = 0.42. An OLS fit between IMAP-DOAS and metered emissions results in $y = 0.98x$ +83 and $R^2$ = 0. 72. The CMF approach in this case significantly underestimated the metered emission rates. In contrast, emission rates derived from IMAP-DOAS showed good correlation and little bias across the population of releases. These values differ slightly from the values published in Rutherford et al., 2023 due to different emission quantification methods and quality filtering however the trends remain the same.





The bias seen in the 2021 CMF result prompted us to revisit the line length assumption described above (i.e., line length pixel minimum must be seven times the length of active bands). A review of our standard line lengths for field campaigns, and specifically the larger regional surveys of the Permian basin performed between 2019-2021 (Cusworth et al., 2019; Cusworth et al., 2022), revealed that the line lengths flown during the 2021 controlled release experiment were an order of magnitude shorter than normal operations (Figure 2). In the CMF formulation (Equation 1), the magnitude of a CH4 enhancement is

directly related to the mean and covariance of pixels contained in a column. With a smaller flightline the column co-variance is calculated with a smaller number of pixels, this means that pixels with a methane enhancement have a larger influence over the co-variance therefore making any deviations from the background (i.e. methane enhancements) possibly smaller. The systematic low bias seen in Figure 1 from the CMF result could be therefore indicative of flight lines that were systematically too short. In contrast, since IMAP-DOAS is a pixel based algorithm and is therefore indifferent to flight lines lengths for

quantification, the much closer agreement to metered emission rates in Figure 1 is additional evidence that short flight lines drove much of the bias in the CMF result.

To provide further evidence that systematically short flight lines bias CMF-derived concentrations (and therefore emissions), we selected a subset of 50 flight lines that were flown during standard campaign operations in the Permian between 2019-

2021. We isolated a single unique plume in each line, cropped the scene around that plume such that it was 1200 pixels (or 6 km) in the along-track direction, and then ran the CMF algorithm. Since the 2019-2021 Permian campaigns were flown at higher altitudes (5-9 km), we cropped these scenes to match the average number of pixels per column during the 2021 controlled release experiment. Therefore, these cropped scenes are representative of the statistical sampling conditions also present in the controlled release CMF results. Figure 3 shows the results of cropping 2019-2021 Permian scenes to the same

pixel dimension as the 2021 CR experiment. What is immediately obvious is that estimated emissions from these cropped scenes are much lower than the standard CMF emission estimates that use all pixels in the along-track direction. An OLS fit between the cropped and standard CMF emissions results in $y = 0.14x + 81$ with $R^2 = 0.47$, showing a severe reduction in estimated emissions. This lends additional evidence that the poor results in Figure 1 were driven primarily by short flight lines.

The analyses described by Figures 1-3 lend confidence that the 2021 bias in CMF results was driven by short flight lines. However, that unfortunately renders the 2021 controlled release experiment incapable of assessing how a properly constrained (i.e., 20-50 km flight line length) CMF algorithm performs quantitatively against a standard metered emission rate. However, given the good performance of IMAP-DOAS on the 2021 controlled release data, we cross-compared emission rates from the 2019-2021 Permian campaigns derived from both CMF and IMAP-DOAS algorithms. This subset includes 60+ plumes that

relate to 20 distinct facilities that were imaged on at least 3 separate days during airborne campaigns by GAO during the Permian 2019-2021 campaigns (Cusworth et al. 2022). These plumes represent a dynamic range of emission rates reported by the CMF algorithm (90 kg/h – 3900 kg/h) and represent a diversity of infrastructure types in this region. The results of the IMAP-DOAS to CMF comparison are shown in Figure 4. The left panel shows instantaneous plume-to-plume emission



comparison for the different retrieval approaches. The data comparison shows general agreement between the two retrieval

approaches. An OLS fit results in $y = 0.72x + 307$ with $R^2 = 0.67$.

In practice, when summarizing the results from campaigns and constructing emission budgets for regions or facilities, we take
persistence-averaged emission rates derived from multiple overpasses of a facility (e.g. the average emission rate over multiple
observations; Cusworth et al. 2021). The right panel of Figure 4 shows the comparison of IMAP-DOAS to CMF after averaging

and applying persistence adjustment to the emissions. Here, the comparison between retrieval approaches shows very close
correspondence: OLS fit: $y = 0.89x + 120$ with $R^2 = 0.82$. As expected, the variability in emissions on a plume-by-plume
basis gets averaged out in Figure 4b. Since the only difference between CMF and IMAP-DOAS derived emission rates are the
concentration retrievals, the improved correlation between single-realization plumes in Figure 4a and multiple-realization
sources in Figure 4b shows that much of the retrieval uncertainty from these two approaches are unbiased because they begin

to converge in agreement with additional sampling.

## 3.2 Matched Filter Sensitivity Tests

The results from Figures 1 and 3 provide evidence that a reduced flight line length hampers the CMF's quantitative
performance however it does not reveal what the appropriate line length is to ensure good quantification. In order to improve
results for the 2022 controlled release and to ensure we are able to accurately quantify plumes operationally we performed an

analysis to determine the minimum line length for good quantification. To do this we identified 8 sources that had at least three
flyovers from GAO and were located in long lines. This resulted in 24 individual scenes to analyze. These scenes ranged from
125 to 27 km long. Using the full flight line lengths, the plumes within each scene have CMF-derived emission rates between
70.7 and 3980 kg/hr. Scenes were acquired from various campaigns including The Permian 2021, Denver-Julesburg Summer
2021, North East 2021, California Summer 2020, and Permian Fall 2021 (A. K. Thorpe et al. 2023; Cusworth et al. 2022).


Each scene was cropped to 2000 pixels (6-10 km) centered on the identified plume. This crop was then iteratively increased
in 1000 pixel (3-5 km) increments until the full scene length was reached. This produced between 6 and 24 cropped images
per scene. Each scene crop was processed through the CMF and an IME algorithm. We used IME over emission rates because
the IME isolated changes in the methane retrieval better than a full emission estimate. Additional details on the methods used

in this analysis can be found in section S2.

We found that in general the IME increases as the pixel count in each column increases, however the rate of increase decreases
with pixel count. While not asymptotic, the small increase in IME after a certain threshold likely means there is an optimal
point (or distance) that is sufficient for quantifying an emission rate. Figure 5 (right panel) shows the pixel count versus the

IME for 3 three different images of one methane plume. The IME increases rapidly at the low pixel counts (short lines) and
then levels off as the line gets longer. We determined this optimal point by calculating the "knee" in the curve or the point





where the change in IME is minimal compared to the change in the scene length. We found that for all the plumes assessed the median "knee" was 7000 pixels or about 21 km when the aircraft is flown as 3 km.  We compared the IME from the 7000 pixels to the standard scene length IME and found generally good agreement (figure 5 left panel). From this we can also
conclude that the minimum line length needed to produce a good quantification with the matched filter is about 21 km. This minimum line length is well within the lengths flown from previous Permian surveys (Figure 2).

### 3.3 2022 Controlled Release

The results from Figure 5 prompted us to require minimum flight line distances of 20 km during the 2022 controlled release experiment. The estimated airborne emission rates compared to metered emissions for the 2022 controlled release are shown
in Figure 6 for both CMF and IMAP-DOAS. An ordinary least squares (OLS) fit between the CMF results and the metered emissions results in y = 0.90x +42, and R²= 0.88. The OLS fit between IMAP-DOAS and metered emissions results $y = 1.14x$ $-19$  and $R^2 = 0.81$. These values differ slightly from the values published in El Abbadi et al., in prep due to different quality filters however the trends remain the same. The results of the CMF are much improved from the 2021 controlled release. These results further highlight the need for appropriately long flightlines. The improved results provide additional assurance that
previous airborne campaigns, most of which have flightlines longer than 20 km, do not have a systematic underestimate.

The CMF and IMAP-DOAS produced comparable results for the 2022 experiment, similar to the results seen from Permian campaigns shown in Figure 4a. However, there exist other features that may corrupt a CMF result, even under long flight lines that were not explicitly tested during the 2022 controlled release experiment. For example, flares or specular reflections from
solar panels often results in saturated or atypical backscattered radiance spectra. If these spectra are observed anywhere in a column and not removed, they can enter into a covariance calculation, causing the CMF results in that column to be biased. In addition, the presence of too many dark pixels, like a water body can also negatively affect the CMF. Though not affected by other pixels, physics-based retrievals like IMAP-DOAS are also prone to retrieval artifacts for complicated pixels. Therefore, controlled release tests across a host of simple to complex observing environments will further refine algorithms to quantify
concentrations and emissions.

### 3.4 Detection Limits

Though biased by short flight lines, the CMF results from the 2021 controlled release experiment performed well at CH4 plume detection. The CMF ability to detect emissions has proven insensitive to line length, however these detection limits are strongly influenced by other observing conditions. To determine detection limits we used logistic regression to build detection
models. For the first set of models (figure 7 left panel) we only used the metered emission rates as the predictor variable and built separate models for the 2021 and 2022 experiments. This allows us to compare the detection limits under two different observing conditions. We also made a more general model (figure 7 right panel), for this we combined both experiments and used wind speed as well as the metered emission rates as predictor variables. Other factors like solar zenith angle and albedo



should also be used as predictor variables however for these experiments there was not enough variability in albedo to use it
as a predictor variable and the solar zenith angle, in this case driven by time of day, was also correlated to the metered release
rates and therefore was also not used as a predictor variable.

In the 2022 controlled release experiment the smallest plume detected was 8.6 kg/hr and the 90% POD was 10 kg/hr. However,
these were ideal conditions. The albedo was 42% in the area surrounding the controlled release site, which is considered high,
and the wind speeds were low (mean 1.78 m/s).  Typical operating conditions are more challenging which can lead higher
minimum detection limits. The 2021 controlled release had more challenging conditions. The wind speeds were higher (mean
2.5 m/s), although not extreme. The surface albedo was 24.5% and also more varied. In the 2021 experiment, the smallest
plume detected was 9.8 kg/hr and the 90% POD was 45 kg/hr. It is important to note that the smallest release rate was also 9.8
kg/hr, so for 2021 there are no data for releases below this value therefore our understanding of detection below 9.8 is
incomplete. When we combine both controlled release experiments and add wind as a predictor variable, we can see more
clearly how detection limits can change depending on condition. The 90% POD ranges from 17 to 68 kg/hr as a function of
wind speed (Figure 7).  These results are consistent with previous controlled release results and with other analysis of POD
curves that showed an AVIRIS-NG 90% POD of 16-33 kg/hr (A. K. Thorpe et al. 2016; Conrad, Tyner, and Johnson 2023).
The POD is also a function of albedo and SZA, however these controlled release experiments were not designed to test and
isolate those parameters.  In practice we anticipate the POD performance to vary across observing regions and seasons.
Specifying and understanding observing conditions is critical to interpreting POD and minimum detection results.

**4 Conclusion**

The non-standard short flight lines flown during the 2021 controlled release experiment resulted in an unexpected low bias in
CMF retrieved $CH_4$ concentrations which propagated into low emission rates. Here, we tested that hypothesis by applying the
pixel-wise IMAP-DOAS retrieval to the 2021 controlled release results. We found a much-improved result, with IMAP-DOAS
derived emission rates showing strong correlation and little bias across the experiment. We subsequently flew longer, more
representative flight lines in the 2022 controlled release experiment and eliminated bias in both the CMF and IMAP-DOAS
retrievals. Both experiments were used to assess the minimum detection limit as well as the 90% probability of detection.
These experiments highlighted the importance of observing conditions when evaluating POD.


Here we also highlight the various strength and weaknesses of the two main retrieval algorithms. The CMF is a fast and reliable
detection algorithm but is sensitive to scene dynamics, specifically scene length, for quantification. Here we show that the
2021 controlled release did not meet scene length requirements for good quantification, but we also show that most carbon
mapper flight lines meet the scene length requirement and produce good quantitative results. IMAP-DOAS on the other hand
is too slow to run as a detection algorithm but can produce more reliable quantification estimates and does not rely on other



aspects of a scene. Ideally, these two algorithms can be used in tandem, the CMF for rapid detection and IMAP-DOAS after for a robust quantification.

As we move towards global monitoring with satellites these results provide confidence in our ability to accurately quantify
methane emissions. These results also highlight the importance of controlled release testing in order to assess and understand retrieval algorithms. As a larger constellation of instruments, and specifically satellites such as EMIT and Planet/Carbon Mapper's Tanager, are used to map methane, more controlled release tests will be needed to fully validate emissions.

**Data availability**

The data from the controlled release experiments are available from the corresponding author upon request. All other data is available at carbonmapper.org/data

**Author contributions**

AKA and DC contributed to the analysis and writing, KO and JF contributed to the matched filter sensitivity tests and IMAP-
DOAS runs respectively. AKT and RD contributed experiment design, advise, and funding.

**Competing interests**

The contact author has declared that neither they nor their co-authors have any competing interests.

**Acknowledgements**

We would like to acknowledge Adam R. Brandt, Philippine M. Burdeau, Yuanlei Chen, Zhenlin Chen, Sahar H. El Abbadi, Jeffrey S. Rutherford, Evan D. Sherwin, and Zhan Zhang for running the controlled release experiments. From the Carbon Mapper team, we would like at acknowledge Kate Howell, David Stepp, Andrew Aubrey, and Ralph Jiorle for supporting the controlled release data analysis. We would like to thank Joseph Heckler and Greg Asner from the GAO team for flight
operations. The Global Airborne Observatory (GAO) is managed by the Center for Global Discovery and Conservation Science at Arizona State University. The GAO is made possible by support from private foundations, visionary individuals, and Arizona State University. Lastly, the Carbon Mapper team acknowledges the support of their sponsors including the High Tide Foundation, Bloomberg Philanthropies, Grantham Foundation, and other philanthropic donors.

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




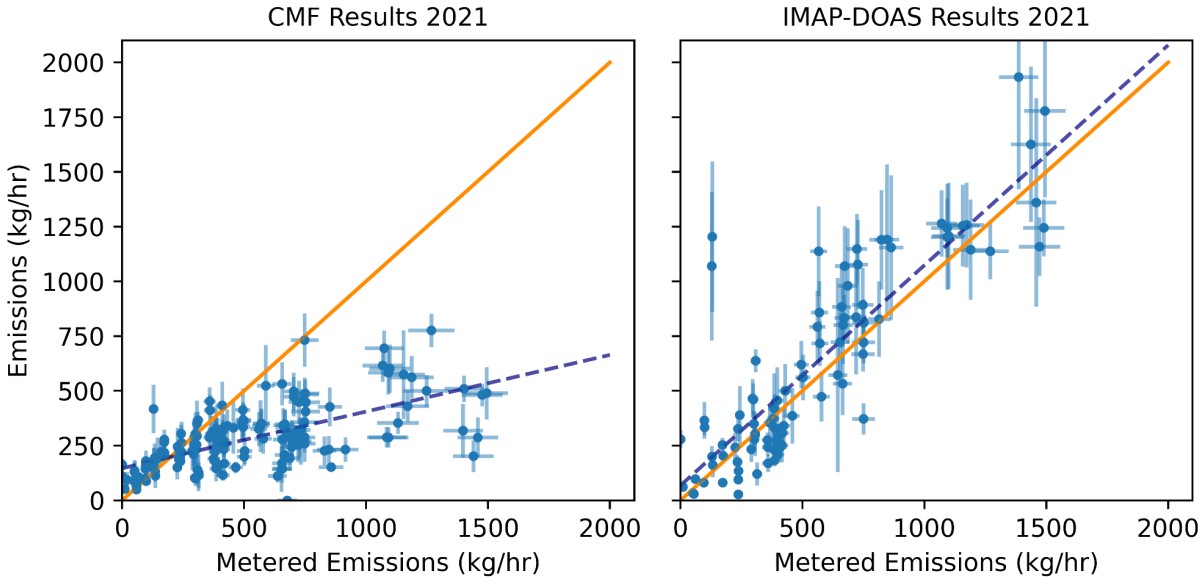


**Figure 1: CMF and IMAP-DOAS comparison to metered emission rates for the 2021 controlled release experiment with shorter than normal flight lines. An OLS fit to CMF results in $y = 0.26x + 146$ with $R^2 = 0.42$. An OLS fit to IMAP-DOAS results in with $y = 0.98x + 83$ and $R^2 = 0.72$. For both figures the solid line represents the 1:1 line and the dashed line is the OLS fit.**

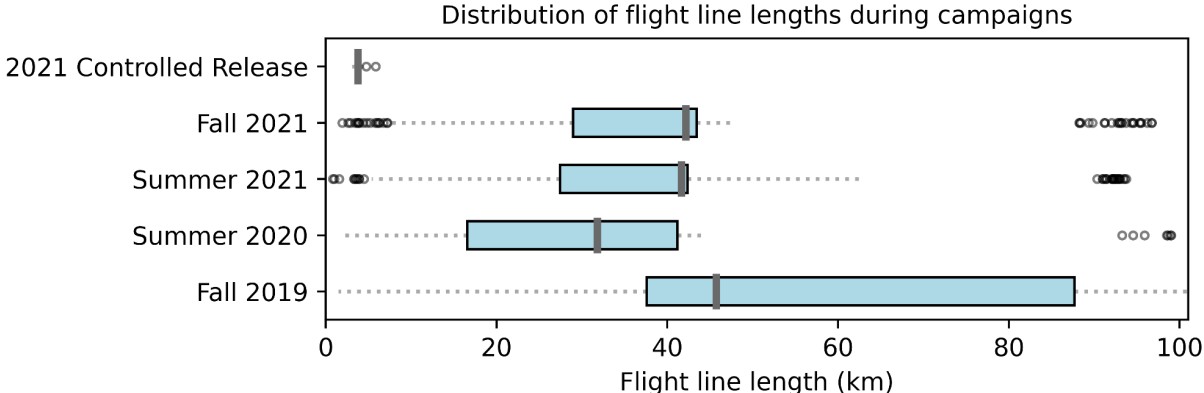

**Figure 2. Flight line lengths during the summer 2021 controlled release experiment compared to Carbon Mapper Permian field campaigns (Cusworth et al., 2021; Cusworth et al., 2022).**



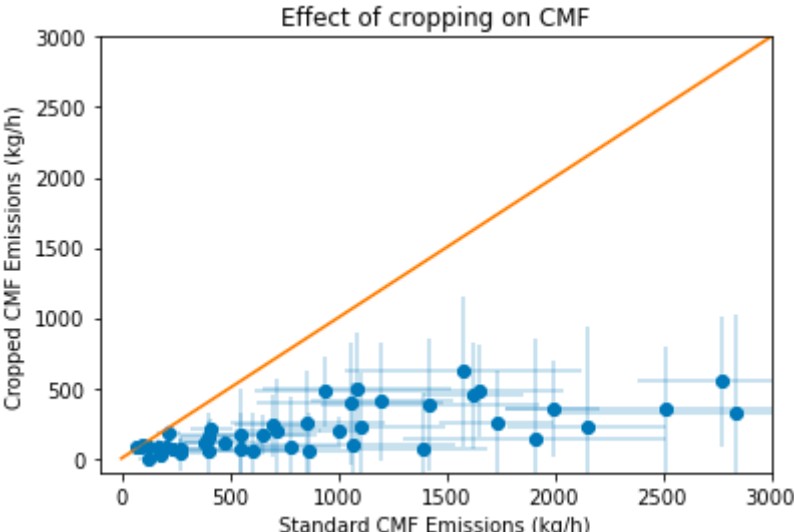

**Figure 3: Effect of cropping scene length on CMF results. Scenes were taken from 2019-2021 Permian campaigns that were flown under normal operations, then cropped to 1200 pixels in length, and the CMF was reran. The resulting emissions are compared to the emissions from the standard full-line CMF processing.**

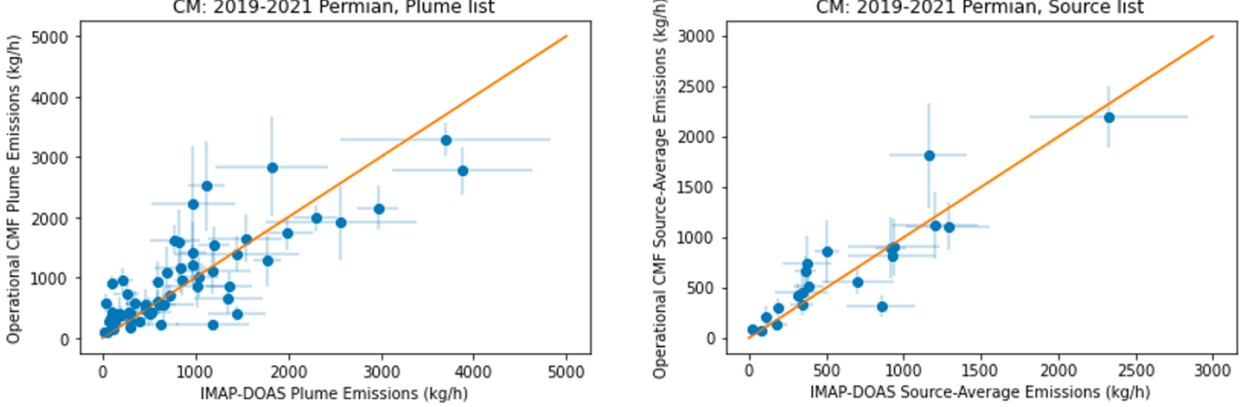

**Figure 4. Comparison of emission rates for a subset of 2019-2021 plumes in the Permian Basin between the operational CMF and IMAP-DOAS. Error bars represent 1-sigma uncertainties on emissions. Regression fits for left panel (plume list with instantaneous emissions): OLS: $y = 0.72x + 307$ with $R^2 = 0.67$.**





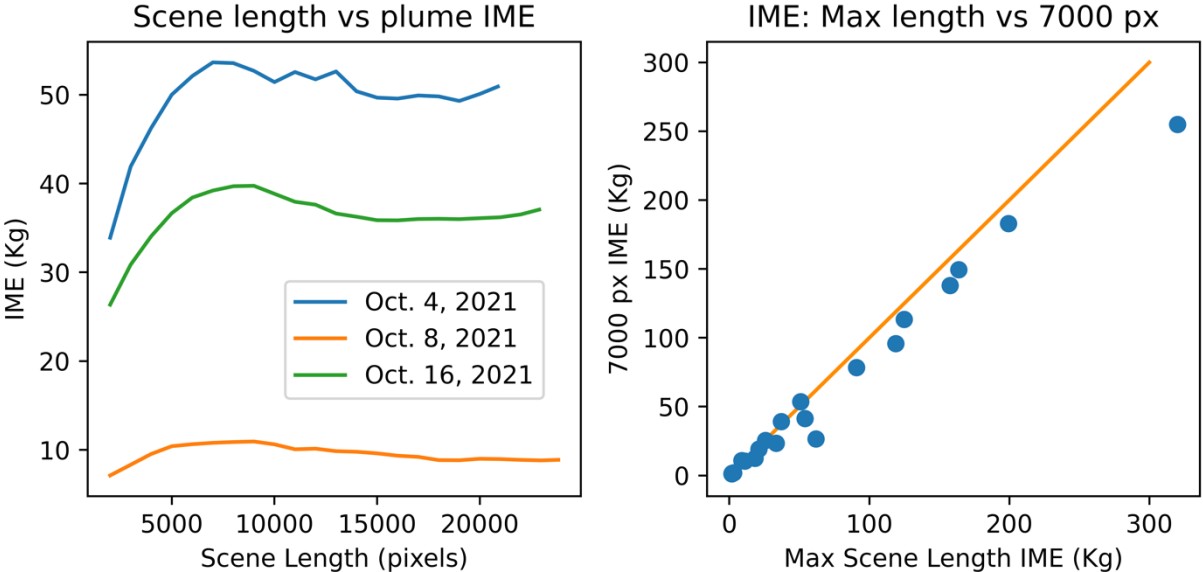

**Figure 5. Left panel shows the scene length (pixels) versus the IME for three plumes from a campaign in October 2021. The right panel shows the comparison of calculated emission rates plumes with a scene length of 7000 pixels (or 21 km when the aircraft is flown at 3 km) compared to the calculated emission rate for the full scene length (length variable depending on scene). The agreement is good, indicating that a ~20 km line is a sufficient length for good quantification.**


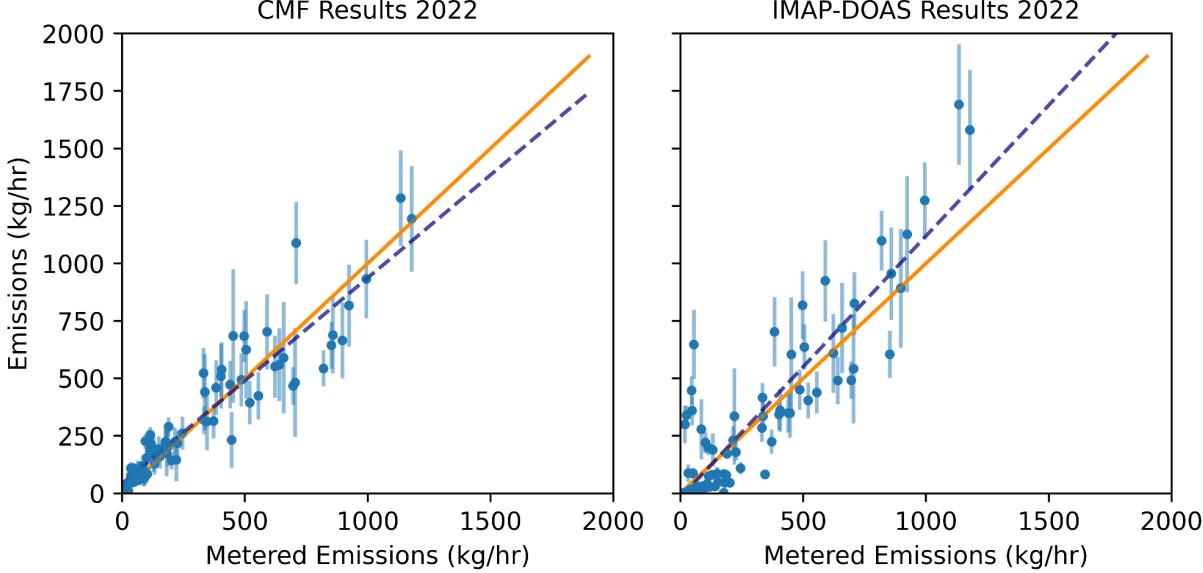

**Figure 6: CMF and IMAP-DOAS comparison to metered emission rates for the 2022 controlled release experiment. An OLS fit to CMF results in $y = 0.90x + 42$ with $R^2 = 0.88$. An OLS fit to IMAP-DOAS results in with $y = 1.14x - 19$ and $R^2 = 0.81$. For both figures the solid line represents the 1:1 line and the dashed line is the OLS fit.**






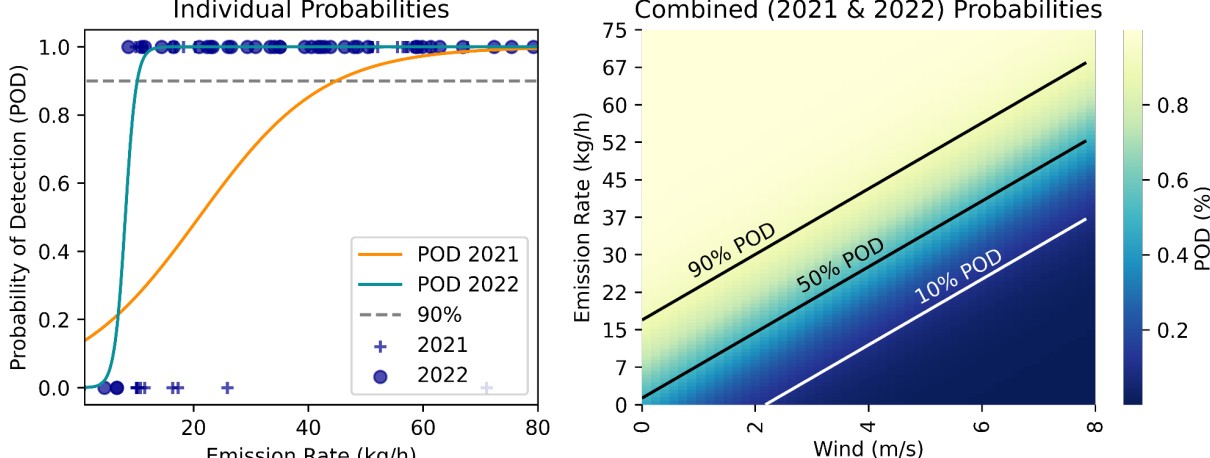

**Figure 7: Probability of Detection (POD) for AVIRIS-NG/GAO instrument using the CMF methane retrieval algorithm. The left plot shows the individual POD curves for the 2021 and 2022 controlled release. The points at zero represent null detects and the points at 1 represent positive detects. The right plot shows the combined POD contours and includes wind.**