# Peer review of "Performance and sensitivity of column-wise and pixel-wise methane retrievals for imaging spectrometers"

_EGUsphere, 2023_

## Referee Comment (RC1)

In the manuscript "Performance and sensitivity of column-wise and pixel-wise methane retrieval for imaging spectrometers", Alana K. Ayasse and colleagues investigate the performance of two different types of algorithms for the retrieval of atmospheric concentration columns of $CH_4$ from spectra acquired by imaging spectrometers. As basis, they not only use observations from the Global Airborne Observatory (GAO), which were collected during two controlled release experiments in 2021 and 2022, but also observations from previous field campaigns. While one of the algorithm (pixel-wise) retrieves $CH_4$ columns for every single spatial pixel by iteratively fitting a simulated spectrum to the measured spectrum (IMAP-DOAS), the second algorithm (column-wise) uses statistics from observed spectra in a flight column (along track) to retrieve $CH_4$ column anomalies (column matched filter, CMF).

The two algorithms compared have their distinct advantages and disadvantages, one being independent from other observations within one flight leg but slow (IMAP-DOAS); the other one depending on a sufficient number of additional observations from the same flight leg but fast (CMF). One key finding of the study is the minimum length of a flight leg required for the fast CMF approach to perform equally well as the slower pixel-wise approach IMAP-DOAS. The authors perform extensive tests and comparisons to find well-suited lengths for different flight legs for the CMF approach. For the comparisons, the retrieved $CH_4$ columns/anomalies are addiationally inverted to fluxes/emissions and they are also compared to the "true" metered emissions. In principle, those findings can be transferred to other imaging instruments and/or similar retrieval approaches. Overall, the manuscript is well-written and conclusive.
The manuscript fits well in the scope of AMT and I recommend publication after some minor modifications along the line of the comments below.

Specific comments:

**P2L40**: Could you provide already here some references for Carbon Mapper and the Global Airborne Observatory (GAO)? Does GAO only consist of an imaging spectrometer or does it comprise an entire suite dedicated to atmospheric measurements? It is also not entirely clear to me how Carbon Mapper and GAO are connected.

**P3L83f**: Are there references describing your standard procedures and the assumption you are putting in more detail (Especially for the definitions of unstable plume morphology and unstable wind conditions)?

**P4L100**: The factor of 7: Has this originated from your long term experience with AVIRIS-ng data or is this based on another study?

**P6L175**: What is the reasoning behind the two different percentiles for the 2021 and 2022 plumes?

**Fig1:** Could you add already in the first Figure that error bars represent 1-sigma uncertainties?

**Fig2**: I would appreciate a little bit more information for the shown graph (Whisker plot?) in the caption in terms of what is 'meant' by the different circles, dots, bars, and lines like the percentiles.

**Fig3**: Would it be possible to add the fitted line and OLS equation as done for Fig. 1?

**Fig4**: Same as for Fig. 3. Would it be possible to add fitted lines to left and right panel and the OLS equations?

**Fig5**: Could you add that the retrieval used for Fig. 5 is CMF in the caption? Additionally, I assume the panels are mixed up. See P8L254 and P9L 259.

In the right panel, it appears that the IME levels-off for larger values. Would you have an explanation for this or is it not significant at all and just a coincidence?

**Fig6**: What happened to the 1-sigma uncertainties for the metered emissions (x-axis)?

**SuppFigS1L26**: What is meant by "… both figues…".

Technical corrections:

**P1L11 and L15:** Could you add that GAO stands for Global Airborne Observatory?

**P1L23, P2L44, P3L87, P4L96L107L110L124, …**: Could you subscript numbers in chemical formulae: CH4 → $CH_4$?

**P4L97-98**: There seems to be a formatting issue regarding new line

**P4L124**: Please remove one "optical depths for".

**P9L259, P9L285L287, …:** Please capitalize "figure" throughout the manuscript.

**P9L263**: "… to require minimum flight…" → "…to required minimum flight…"?

**SuppP2L33**: Typo: Covid CA 202

**SuppFigS1L25**: Typo: Duren e al 2019

---

## Author Comment (AC1)

Thank you to RC2 for putting time and effort to read and review our manuscript. The reviews were helpful and insightful and have made the paper stronger. Below are the comments from RC2 and the responses by the authors are in blue.

**RC2**: 'Comment on egusphere-2023-1420', Anonymous Referee #2

The manuscript by Ayasse et al. departs from the significant difference obtained in the methane emission estimation results during two controlled emission experiments in order to understand the cause of the flux underestimation obtained in the first campaign, compare the results obtained by processing the data with two different algorithms (CMF and IMAP-DOAS) and, therefore, better understand the behavior of both algorithms and give a guide to which algorithm is more appropriate for each case. In addition, it suggests a solution to avoid obtaining underestimates from flight campaigns if the Matched Filter approach is to be used to process the data. The study has been done with data collected with the GAO sensor in the controlled release experiments organized by Standford University in 2021 and 2022 and also in previous campaigns performed in other areas of the United States, which adds robustness to the study.

I consider the methodology used in this work appropriate and the analysis of the results rigorous and valuable to the scientific community. In addition, the manuscript reads well and is easy to follow. However, I would like to see a little more elaboration on some of the points I list below and some issues need to be addressed. I recommend the publication of this article once the following points are corrected or taken into consideration:

**Major comments:**

The abstract mentions that the column-wise retrieval algorithm is sensitive to the flight line length so that it can have a systematic low bias on short flight lines but does not have to be on long enough flights and that this bias is not present in the per-pixel retrieval. Reading this in the abstract, one might first think, why not always use per-pixel retrieval and avoid bias? I suggest including a line clarifying that, on the other hand, per-pixel retrieval has a much higher computational cost, so for large amounts of data processing, it is more optimal to use column retrieval.

The abstract has been edited to include the following sentence "However, the pixel-wise retrieval is computationally expensive and the column-wise retrieval algorithms can produce good results when the flight line length is sufficiently long."

In the same way, in the text, it is mentioned that CMF is computationally more efficient than IMAP-DOAS, but I miss a more quantitative comparison of this difference to help readers better understand the difference. I think it would be helpful to add a sentence, for example, in the methodology section, with an indicative example of "for an image of x length/ xx number of pixels, processing with CMF would take about x minutes, while with IMAP-DOAS xx hours" or similar.

P5 L192 has been edited to read "However, the current processing time of IMAP-DOAS makes its operational use limited. At best it takes 1 second per pixel to run, therefore 5,700 pixels (which typically is a 300 by 300 m area) can take 2-3 hours to run. In contrast it takes about 7 minutes to run an entire 3.3 million pixel scene with the CMF. Future processing improvements may significantly reduce the computation cost, but it is unlikely IMAP-DOAS will ever be as computationally efficient as the CMF."

Lines 169-176: here, I understand that for the CMF, you first remove the background and then obtain the mask of the plume with the values that have remained above the removed background, and in IMEP-DOAS, you first have the mask and then with the values that are not included in the mask you determine the concentration of the background. If correct, how do you calculate the initial mask with the IMAP-DOAS method? Do you do it manually? Taking first an indicative background value? I would appreciate a clarification on this in the text.

The CMF produces a methane enhancement above background as its output. P4 L125 and line P7 L215 have both been edited to make this clearer. P7 L215 now reads "…the CMF (which retrieves an enhancement above background and therefore does not have background methane incorporated in the pixel values)…".

Given the noise in the image it is still necessary to threshold the pixels in order to identify pixels associated with the plume, however there is no additional adjustment (such as subtracting off background values) needed to the data. For IMAP-DOAS, the algorithm retrieves a total column concentration, which means that included in the per-pixel value is the background concentration. We ultimately want the enhancement above background, so we need to subtract off the background concentration. To do that we first identify the plume using the methods described in P6 L215-217 ("we calculated thresholds using the 80-98th percentile of 1 km area around the plume origin. The thresholds were then used to filter out low values, we then calculated connected components starting at the plume origin, then used a dilation to fill in gaps.")

Line 305: regarding the sentence "In practice, we anticipate the POD performance to vary across observing regions and seasons.", this was already shown in Gorroño et al., 2023 https://amt.copernicus.org/articles/16/89/2023/amt-16-89-2023.html with Sentinel 2, which it would be worth mentioning.

A citation to Gorrono et al 2023 has been added here.

Lines 324-325: the first sentence of the paragraph sounds a bit out of place. Satellites typically have a specific and invariable swath and do not take longer or shorter images. However, it may happen to get a very variable image with, for example, a large presence of water or a higher methane content than normal, preventing an appropriate formation of the covariance matrix and leading to under- or over-estimates. This is a discussion that I missed in the manuscript and could fit here to make sense of the sentence.

The purpose here is to stress how important it is to use controlled releases to test, evaluate, and understand how accurately we can quantify point source methane emission. In referencing satellites, the point is to show that this type of analysis will become more critical once routine

global measurements are used for applications beyond science. I have edited the final paragraph to be less focused on satellites are more focused on global and routine measurements with remote sensing in general.

Line 326: about the sentence "As a larger constellation of instruments, and specifically satellites such as EMIT and Planet/Carbon Mapper's Tanager, are used to map methane, ...", first, EMIT is not a satellite and second, why "specifically" EMIT and Planet/Carbon Mapper? If the reason is the common use of the Matched Filter to process the data, generally in studies with PRISMA, EnMAP, or Gaofen5, the CMF is also used to optimize the computational cost (e.g. Irakulis-Loitxate et al. 2021 https://www.science.org/doi/epdf/10.1126/sciadv.abf4507, Guanter et al. 2021 https://www.sciencedirect.com/science/article/abs/pii/S0034425721003916, Roger et al. 2023 https://eartharxiv.org/repository/view/5235/, Nesme et al. 2021 https://doi.org/10.3390/rs13244992). I suggest changing the sentence by removing the "and specifically satellites such as EMIT and Planet/Carbon Mapper's Tanager," and adding the reference of Jacob et al., 2022 after "... constellation of instruments", or simply removing "specifically satellites" and name the other satellites as well.

I removed mention to specific satellites. The final sentence read " As a larger constellation of instruments are used to map CH4 (Jacob et al., 2022) , more controlled release tests will be needed to fully validate emissions."

References section: the references Ayasse et al., Foote et al., Maasakkers et al., and Ocko et al. are not mentioned in the text.

The references have been edited.

**Minor corrections:**

Line 21 and 22: in addition to the references you cite there, nowadays, there are quite a lot of studies confirming that a significant component of the anthropogenic methane budget comes from a relatively small population of high emission point sources (e.g. Frenkenberg et al., 2016 PNAS, Irakulis-Loitxate et al., 2022 ES&T and ES&TL, Ehret et al., 2022 ES&T), so, for correctness, I suggest you put an "e.g." at the beginning of the list.

The edit has been made as suggested.

Line 24: at the end of the sentence " ... effects in the next few decades.", I miss a reference that affirms the sentence. Reference to Ocko et al., 2021 would be appropriate here, which is listed in the references but not mentioned in the manuscript. Otherwise, reference to IPCC, 2023: Summary for Policymakers. doi: 10.59327/IPCC/AR6-9789291691647.001would also be appropriate.

The edit has been made as suggested.

Line 26: I think it is not totally fair to say that airborne imaging spectrometers can repeatedly map large areas. A satellite does have the ability to map large areas every few days over long periods of time (e.g., Irakulis-Loitxate et al., 2022 https://pubs.acs.org/doi/full/10.1021/acs.est.1c04873), but airborne mapping is limited to campaign periods. It is true that in the same campaign, the same locations can be mapped several times, but then there will not be a revisit, in the best cases, before ~one year. I suggest changing the sentence to "In particular, airborne imaging spectrometers with shortwave infrared (SWIR) sensitivity have emerged as useful tools for point source quantification due to their high spatial resolution, low detection limit, and ability to map large areas for point sources" or similar.

*The edit has been made as suggested.*

Line 29: at the end of the sentence "... environmental variables including surface illumination and atmospheric transport.", a reference is missing. An appropriate reference would be Gorroño et al., 2023 https://amt.copernicus.org/articles/16/89/2023/amt-16-89-2023.html

*The reference has been added as suggested.*

Line 31: for the sentence "... and complexity.", please add a reference, e.g., Jacob et al., 2022 https://acp.copernicus.org/articles/22/9617/2022/acp-22-9617-2022.html

*The reference has been added as suggested.*

Line 62: I think here you are referring to this other paper by Sherwin et al. https://www.nature.com/articles/s41598-023-30761-2 not listed in the references.

*The references have been edited.*

Lines 74 and 79: add space between 3 and km.

*The edit has been made as suggested.*

Line 96: Please add a reference at the end of the sentence "... $CH_4$ has known absorption properties.". The reference of your paper, Ayasse et al., 2018, listed in the bibliography (but not mentioned in the text) would work great.

*The references have been edited.*

Line 97, 176, 260: for consistency, matched filter => CMF. It also looks like there is a formatting error on this line.

*The edits have been made as suggested.*

Line 98, 101, 196: for consistency, methane => $CH_4$

*The edits have been made as suggested.*

Line 112, 147, 244, 303: for consistency with the rest of the citations, Thorpe et al., 2017 (without the name/acronyms).

*The edit has been made as suggested.*

Line 144: add Where before So to make it easier to read.

*The edit has been made as suggested.*

Line 155: again, for consistency with other citations, Frankenberg et al. 2016 (without the name)

*The edit has been made as suggested.*

Line 213: for clarity, add "CMF results" or "left panel" after Figure 1.

*The edit has been made as suggested.*

Lines 232-234 and 273/Figure 4: add a) and b) in the panels of the figure, or change the test to Figure 4 left and right.

*Right and left panel have been added.*

Line 250: section S2 => Section 2 or Methodology section

*This is a reference to the supplemental section and this has been changed in the text for clarity.*

Line 259, 285, 287: the F in capital letter in "figure 5" and "figure 7"

*The edits have been made as suggested.*

Line 293: POD should also be defined in the test

*The edit has been made as suggested.*

Line 304: Solar Zenit Angle (SZA)

*The edit has been made as suggested.*

Figure 2: for completeness, I think it is worth including in this figure the flight line length of the second controlled release experiment in 2022.

*Figure 2 has been edited to include the 2022 controlled release.*

Figure 2: in the figure caption, it should be explained what the gray bar and black circles are.

For figure 2 the caption has been edited to include " The data is displayed as a box plot with the blue box representing the inter quartile range, the gray bar is the median, the dashed lines are the min and max, and the black circles are outliers"

Figure 4: in the title of the plots, what does CM mean? If it refers to Comparison of Methods, I suggest removing it.

CM has been removed from the figure titles.

---

## Author Comment (AC2)

Thank you to RC1 for putting time and effort to read and review our manuscript. The reviews were helpful and insightful and have made the paper stronger. Below are the comments from RC1 and the responses by the authors are in blue.

**RC1**: 'Comment on egusphere-2023-1420', Anonymous Referee #1

In the manuscript "Performance and sensitivity of column-wise and pixel-wise methane retrieval for imaging spectrometers", Alana K. Ayasse and colleagues investigate the performance of two different types of algorithms for the retrieval of atmospheric concentration columns of CH4 from spectra acquired by imaging spectrometers. As basis, they not only use observations from the Global Airborne Observatory (GAO), which were collected during two controlled release experiments in 2021 and 2022, but also observations from previous field campaigns. While one of the algorithm (pixel-wise) retrieves CH4 columns for every single spatial pixel by iteratively fitting a simulated spectrum to the measured spectrum (IMAP-DOAS), the second algorithm (column-wise) uses statistics from observed spectra in a flight column (along track) to retrieve CH4 column anomalies (column matched filter, CMF).

The two algorithms compared have their distinct advantages and disadvantages, one being independent from other observations within one flight leg but slow (IMAP-DOAS); the other one depending on a sufficient number of additional observations from the same flight leg but fast (CMF). One key finding of the study is the minimum length of a flight leg required for the fast CMF approach to perform equally well as the slower pixel-wise approach IMAP-DOAS. The authors perform extensive tests and comparisons to find well-suited lengths for different flight legs for the CMF approach. For the comparisons, the retrieved CH4 columns/anomalies are additionally inverted to fluxes/emissions and they are also compared to the "true" metered emissions. In principle, those findings can be transferred to other imaging instruments and/or similar retrieval approaches. Overall, the manuscript is well-written and conclusive. The manuscript fits well in the scope of AMT and I recommend publication after some minor modifications along the line of the comments below.

**Specific comments:**
P2L40: Could you provide already here some references for Carbon Mapper and the Global Airborne Observatory (GAO)? Does GAO only consist of an imaging spectrometer or does it comprise an entire suite dedicated to atmospheric measurements? It is also not entirely clear to me how Carbon Mapper and GAO are connected.

Line 44 has been edited to read. 'Carbon Mapper, a non-profit that provides facility scale methane emission data via remote sensing, participated in both experiments. Carbon Mapper contracted the Global Airborne Observatory (GAO) imaging spectrometer, which has the same design as AVIRIS-NG, to collect the raw radiance data which Carbon Mapper then processed to emission estimates.'

To clarify further, GAO is an airborne platform, The main instrument on it is the imaging spectrometer, it is also equipped with a high-resolution camera and a lidar, however neither of those were used for the research conducted in this paper. Carbon Mapper is a non-profit research

group that has expertise in using imaging spectrometer data to detect and quantify methane plumes. We contract 2 airborne imaging spectrometers that are essentially identical. One is AVIRIS-NG operated by NASA JPL the other is GAO operated by ASU. For these experiments we contracted GAO.

P3L83f: Are there references describing your standard procedures and the assumption you are putting in more detail (Especially for the definitions of unstable plume morphology and unstable wind conditions)?

A section has been added to the supplement section that describes what we looked for when we say unstable wind or plume morphology.

P4L100: The factor of 7: Has this originated from your long term experience with AVIRIS-ng data or is this based on another study?

This is based on our experience.

P6L175: What is the reasoning behind the two different percentiles for the 2021 and 2022 plumes?

Quantifying a concentration background is a core challenge for total column retrieval methods. Continuing work is needed to quantify backgrounds dynamically given a diverse set of observing conditions. We have edited the document to make this more clear. P11L335: However, continuing work is needed to robustly quantify background concentrations over a diverse set of observing conditions.

Fig1: Could you add already in the first Figure that error bars represent 1-sigma uncertainties?

The figure 1 caption has been edited to say the error bars represent 1 sigma uncertainties.

Fig2: I would appreciate a little bit more information for the shown graph (Whisker plot?) in the caption in terms of what is 'meant' by the different circles, dots, bars, and lines like the percentiles.

For figure 2 the caption has been edited to include " The data is displayed as a box plot with the blue box representing the inter quartile range, the gray bar is the median, the dashed lines are the min and max, and the black circles are outliers"

Fig3: Would it be possible to add the fitted line and OLS equation as done for Fig. 1?
Fig4: Same as for Fig. 3. Would it be possible to add fitted lines to left and right panel and the OLS equations?

Fit lines have been added to both figure 3 and 4.

Fig5: Could you add that the retrieval used for Fig. 5 is CMF in the caption? Additionally, I assume the panels are mixed up. See P8L254 and P9L 259. In the right panel, it appears that the IME levels-off for larger values. Would you have an explanation for this or is it not significant at all and just a coincidence?

The figure caption has been edited to include retrieval and the panels listed in the text were indeed mixed up. The leveling off at higher line length is expected and then explanation for this is discussed in P7 L 250-255.

Fig6: What happened to the 1-sigma uncertainties for the metered emissions (x-axis)?

1-sigma uncertainties are added for the x-axis, they are just so small it is hard to see them.

SuppFigS1L26: What is meant by "… both figues...".

This has been edited.

**Technical corrections:**
P1L11 and L15: Could you add that GAO stands for Global Airborne Observatory?

The acronym has been removed from the abstract and it is defined in the introduction on L47.

P1L23, P2L44, P3L87, P4L96L107L110L124, …: Could you subscript numbers in chemical formulae: CH4 CH4?

We will reformat according to AMT guidelines if accepted for publication.

P4L97-98: There seems to be a formatting issue regarding new line P4L124: Please remove one "optical depths for".

One has been removed.

P9L259, P9L285L287, …: Please capitalize "figure" throughout the manuscript.

Figure has been capitalized throughout the manuscript.

P9L263: "… to require minimum flight…" to "…to required minimum flight…"?

Edited to make better grammatical sense.

SuppP2L33: Typo: Covid CA 202

Changed in text.

SuppFigS1L25: Typo: Duren e al 2019 responses

Changed in text.

---

## Author Response (AR2)

Please undertake the following technical corrections before publication:

Line 83 of manuscript version 3: add space between 3 and km

Line 313: Solar Zenith Angle

The above two changes were made in the manuscript.

Fig 3: add the OLS fit equation (as in Fig 1) following the request of referee 1

The OLS equation has been added to the figure caption.

Fig 4, left panel: the OLS fit line seems not to fit

A mistake was made when making the figure, The OLS line was incorrect and has been corrected.

Fig 6: mention in the caption that the 1-sigma uncertainties on the x-axis are too small to see

Supplement, end of line 18: remove "in"

The above two changes were made in the manuscript.